# miR-22 Modulates Lenalidomide Activity by Counteracting MYC Addiction in Multiple Myeloma

**DOI:** 10.3390/cancers13174365

**Published:** 2021-08-29

**Authors:** Daniele Caracciolo, Caterina Riillo, Giada Juli, Francesca Scionti, Katia Todoerti, Nicoletta Polerà, Katia Grillone, Lucia Fiorillo, Mariamena Arbitrio, Maria Teresa Di Martino, Antonino Neri, Pierosandro Tagliaferri, Pierfrancesco Tassone

**Affiliations:** 1Department of Experimental and Clinical Medicine, Magna Græcia University, 88100 Catanzaro, Italy; d.caracciolo@unicz.it (D.C.); caterina.riillo1@studenti.unicz.it (C.R.); giadajuli@libero.it (G.J.); nicoletta.polera@studenti.unicz.it (N.P.); k.grillone@unicz.it (K.G.); luciafiorillo@unicz.it (L.F.); teresadm@unicz.it (M.T.D.M.); tagliaferri@unicz.it (P.T.); 2Institute of Research and Biomedical Innovation (IRIB), Italian National Council (CNR), 98164 Messina, Italy; scionti@unicz.it; 3Hematology, Fondazione Cà Granda IRCCS Policlinico, 20019 Milan, Italy; katia.todoerti@policlinico.mi.it (K.T.); antonino.neri@unimi.it (A.N.); 4Institute of Research and Biomedical Innovation (IRIB), Italian National Council (CNR), 88100 Catanzaro, Italy; mariamena.arbitrio@irib.cnr.it; 5Department of Oncology and Hemato-Oncology, University of Milan, 20019 Milan, Italy

**Keywords:** multiple myeloma, MYC, microRNA, miR-22, lenalidomide, NK cells

## Abstract

**Simple Summary:**

MYC-driven deregulation of microRNAs represents a critical event in human malignancies, including multiple myeloma (MM). Although the introduction of new therapeutic strategies has prolonged survival of patients, MM remains an incurable disease, often due to the onset of drug resistance. MYC hyperactivation is involved in the development of resistance to immunomodulatory imide drugs (IMiDs), but the mechanism is still unclear. Here, we report that MYC represses the transcription of tumor suppressor miR-22 in MM, and that low miR-22 expression is associated with IMiD resistance in MM patients. By in silico and in vitro analysis, we show that miR-22 mimics affect MYC signaling, leading to MM cell death in MYC proficient cells. Furthermore, we demonstrate here that lenalidomide treatment enhances miR-22 activity by reducing the MYC inhibitory effect, and that the combination of lenalidomide with miR-22 mimics restores drug sensitivity, leading to synergistic anti-MM activity.

**Abstract:**

*Background:* MYC is a master regulator of multiple myeloma (MM) by orchestrating several pro-tumoral pathways, including reprograming of the miRNA transcriptome. MYC is also involved in the acquirement of resistance to anti-MM drugs, including immunomodulatory imide drugs (IMiDs). *Methods:* In silico analysis was performed on MM proprietary and on public MMRF-CoMMpass datasets. Western blot and chromatin immunoprecipitation (ChIP) experiments were performed to validate miR-22 repression induced by MYC. Cell viability and apoptosis assays were used to evaluate lenalidomide sensitization after miR-22 overexpression. *Results:* We found an inverse correlation between MYC and miR-22 expression, which is associated with poor outcome in IMiD-treated MM patients. Mechanistically, we showed that MYC represses transcription of miR-22, which, in turn, targets MYC, thus establishing a feed-forward loop. Interestingly, we found that IMiD lenalidomide increases miR-22 expression by reducing MYC repression and, most importantly, that the combination of lenalidomide with miR-22 mimics results in a synergistic direct and NK-mediated cytotoxic activity. *Conclusions: Taken* together, our findings indicate that: (1) low miR-22 expression could represent a potential predictive biomarker of poor lenalidomide response in MM patients; and (2) miR-22 reduces MYC oncogenic activity, thus triggering a novel synthetic lethality loop, which sensitizes MM cells to lenalidomide.

## 1. Introduction

Multiple myeloma (MM) is a hematologic malignancy characterized by clonal proliferation of malignant plasma cells in the bone marrow (BM), associated with bone disease, serum monoclonal gammopathy, immune suppression, and end-organ damage [1]. Despite the recent relevant advancement in the treatment of MM by new drugs, which have significantly improved patients’ outcome, MM is still an incurable malignancy [2]. Therefore, a deeper molecular understanding of its pathogenesis and the investigation of crucial nodes of MM vulnerability are needed to develop novel therapeutic strategies against still undruggable critical targets, such as MYC [3]. Indeed, MYC is overexpressed during MM progression, with increases in expression, as observed during MGUS to MM transition, and with the highest levels of expression in plasma cell leukemia (PCL) [4,5]. Furthermore, MYC plays a critical role in drug resistance to conventional anti-MM therapy, such as melphalan [6], as well as to immunomodulatory imide drugs (IMIDs). In particular, a recent study showed a significant upregulation of MYC protein expression at the time of lenalidomide-refractory disease, as compared to patients at the time of diagnosis, suggesting that increases in MYC strongly contribute to the development of lenalidomide resistance [7].

From a mechanistic point of view, MYC is an essential regulator of cancer cell growth, since it orchestrates a potent pro-cancer program across multiple cellular pathways [8,9], steadily including the transcription regulation of microRNAs (miRNAs) [10]. These are small noncoding RNAs of 19–25 nucleotides that regulate the gene expression by degrading or inhibiting the translation of target mRNAs by 3′UTR binding [11]. Emerging preclinical evidence indicates the targeting of deregulated miRNAs in cancer cells as a promising therapeutic option [12,13,14,15,16], and advanced studies [17,18,19] have already approached to early clinical trials in patients with refractory advanced malignancies, including MM (2017-002615-33, EudraCT Number).

Interestingly, the predominant influence of MYC on miRNA expression is a widespread downregulation, indicating that miRNA repression promotes MYC-mediated tumorigenesis and drug resistance acquirement [20]. In this scenario, while several experimental data demonstrated a critical role of specific miRNAs in mediating bortezomib [21], melphalan [22], and dexamethasone [23] effects, little is known about the causal role of miRNAs in IMiDs’ anti-MM activity [24].

On these premises, we here investigate the transcriptional suppressive activity of MYC on miRNA expression in MM patients, and we disclose a novel MYC–miR-22 feed-forward loop of relevance in the regulation of MM cell survival and sensitivity to IMiD lenalidomide.

## 2. Materials and Methods

### 2.1. Multiple Myeloma Cell Lines, Primary Cells, and Reagents

Peripheral blood mononuclear cells (PBMCs) and primary cells from multiple myeloma patient bone marrow aspirates were isolated using Ficoll–Hypaque density gradient sedimentation, and then separated by antibody-mediated selection (Miltenyi Biotec). Multiple myeloma cell purity was assessed by flow cytometry analysis using a phycoerythrin-conjugated CD138 monoclonal antibody. Multiple myeloma cell lines (HMCLs), RPMI-8226 and OPM2, were purchased from ATCC (Manassas, VA, USA). AMO1, U266 were purchased from DSMZ (Braunschweig, Germany). AMO1 bortezomib-resistant (ABZB) were kindly provided by Dr. Christoph Driessen (Eberhand Karls University, Tübingen, Germany). Multiple myeloma cells were cultured in RPMI1640 (Thermo Fisher Scientific, Waltham, Massachusetts, USA) supplemented with 10% or 20% (U266 and primary MM cells) of FBS (Lonza Group, Basel, Switzerland). INA-6 cell line (kindly provided by Dr. Renate Burger, University of Erlangen-Nuremberg, Erlangen/Nuremberg, Germany) was cultured in the presence of recombinant human IL6 (2.5 ng/mL, R&D Systems, Minneapolis, MN, USA). OPM2 and their isogenic lenalidomide-resistant cells (OPM2LenRes) were kindly provided by Dr. Yuan Xiao Zhu (Division of Hematology, Mayo Clinic, Scottsdale, AZ, USA). All these cell lines were immediately frozen and used from the original stock within 6 months. Lenalidomide (SML2283), 10058-F4 (F3680), and JQ1 (SML0974) were purchased from Merk.

### 2.2. Analysis of Cell Viability and Apoptosis

Cell viability was analyzed by Cell Titer-Glo (Promega); apoptosis was evaluated by flow cytometric (FACS) analysis following annexin V-7AAD staining (BD Pharmingen).

### 2.3. Flow Cytometry and Degranulation Assay

The expression of the NKG2D and DNAM-1 ligands on different MM cells was evaluated, after 24 h from transfection with miR-22 or miR-NC, by using fluorochrome-conjugated antibodies against MIC-A/B (Becton Dickinson, Franklin Lakes, NJ, USA), ULBP 1 (R&D Systems), ULBP 2-5-6 (R&D Systems), and CD47 (Becton Dickinson) according to the producer’s guidelines. NK cell degranulation was evaluated using CD107a staining. Specifically, miR-22- or miR-NC-transfected MM cell lines were washed twice in complete medium and incubated with PBMCs in an effector–target ratio of 10:1 in a U-bottom 96-well plate in complete medium at 37 °C and 5% CO_2_ in the presence of anti-CD107a/PE (Becton Dickinson) for 3 h. Cells were then stained with anti-CD3/PcP and anti-CD56/APC to identify NK cell population. NK cells positive for CD107a were considered as degranulating/activated cells able to induce cytotoxicity, as previously described [25]. All experiments were acquired by an ATTUNE Nxt (Thermo Fisher Scientific, Waltham, Massachusetts, USA) flow cytometer. For each sample, at least 1 × 10^4^ events in the gate of interest were acquired.

### 2.4. Cytotoxicity Assay

MM target cells were labeled with FAR-RED (Invitrogen) viable marker, according to the manufacturer’s instructions, and then were transfected with miR-22 or scrambled control. After 24 h, transfected cells were cocultured with human healthy donors-derived PBMCs at different effector/target ratios for 24–48 h at 37 °C and 5% CO_2_, and then stained with 7ADD (Becton Dickinson). Cytotoxicity was detected by flow cytometry (Attune NxT Flow cytometer, Thermo Fisher Scientific) as the percentage of 7AAD+/FAR RED+ cells, as well as by LDH release assay (CyQUANT™ LDH Cytotoxicity Assay, Thermo Fisher Scientific, Waltham, Massachusetts, USA), according to the manufacturer’s instructions.

### 2.5. Transduction of Cells

To generate cells stably expressing c-MYC, U266 were transduced with Precision LentiORF human MYC (GE Dharmacon, Lafayette, CO, USA).

### 2.6. RNA Extraction and Quantitative Real-Time PCR

Total RNA extraction from MM cells and qRT-PCR were performed as previously described [26]. Briefly, total RNA was extracted from cells using TRIzol^®^ reagent (Gibco, Life Technologies, Carlsbad, CA, USA), following the manufacturer’s instructions. The RNA quantity and quality was assessed through NanoDrop^®^ (ND-1000 Spectrophotometer). To evaluate gene expression levels, 1000 ng of total RNA was reverse transcribed to cDNA using the “High Capacity cDNA Reverse Transcription Kit” (Applied Biosystems, Carlsbad, CA). The single-tube TaqMan assay (Applied Biosystems, Carlsbad, CA, USA) was used to detect and quantify MYC (Hs00153408_m1), according to the manufacturer’s instructions, using Viia 7 Dx multicolor detection system (Applied Biosystems, Carlsbad, CA, USA). The obtained threshold cycle (CT) values were normalized on GAPDH (Hs03929097_g1). Comparative real-time polymerase chain reaction (RT-PCR) was carried out in triplicate, including no-template controls. Relative expression was calculated using the comparative cross threshold (Ct) method. Taq-Man^®^ MicroRNA assays (Life Technologies) were used to detect and quantify mature mir-22-3p (assay ID 000398), according to the manufacturer’s guidelines on a ViiA7 System (Thermo Fisher Scientific, Waltham, Massachusetts, USA). MiR-22-3p expression was normalized on RNU44 (assay ID 001094).

### 2.7. Gene Expression Profiling

Highly purified PC samples (CD138 ≥ 90%) from the bone marrow of 129 multiple myeloma (MM), 36 plasma cell leukemia (PCL), and 20 relapsed patients, together with 4 healthy donors (N) were profiled on the GeneChip^®^ Human Gene 1.0 ST array (Affymetrix, Santa Clara, CA) [27]. A subset of them (97 MM, 30 PCL, 13 relapsed cases, and 4 N) was also profiled on GeneChip^®^ miRNA 3.0 arrays (Affymetrix, Santa Clara, CA, USA). The GEP data have been deposited in the NCBI Gene Expression Omnibus database (GEO; http//www.ncbi.nlm.nih.gov/geo (accessed on 10 July 2021); accession No. GSE66293, GSE73452 and GSE70254).

### 2.8. In Vitro Transfection of MM Cells

Synthetic *mir*VanaTM miR-22-3p mimic and inhibitor were purchased from InvitrogenTM (Thermo Scientific). All the oligos were used at 100 nmol/L final concentration. MM cells were transfected using Neon Transfection System (Invitrogen™) (2 pulses at 1.050 V, 30 ms).

### 2.9. Luciferase Reporter Experiments

The wild-type and mutated 3′ untranslated region (UTR) of MYC (NM_002467.6) was cloned in pEZX-MT05 (GLuc/SEAP) vector and purchased from GeneCopoeia (Rockville, MD, USA). Multiple myeloma cells were electroporated using 2.5 μg of each plasmid; for each plate, 100 nmol/L of the synthetic miR-22 or miR-NC was used. Secreted Gaussia luciferase (GLuc) and secreted alkaline phosphatase (SEAP), which allows normalization of GLuc activity, were analyzed by Secrete-Pair™ Dual Luminescence Assay Kit (GeneCopoeia, Rockville, MD, USA).

### 2.10. Western Blot Analysis

Whole-cell protein extracts were prepared from MM cells and from PBMCs in NP40 Cell Lysis Buffer (Novex^®^) containing a cocktail of protease inhibitors (Sigma, Steinheim, Germany). Cell lysates were loaded and PAGE separated. Proteins were transferred by Trans-Blot^®^ TurboTM Transfer Starter System for 7 min. After protein transfer, the membranes were blotted with antibodies listed in the table and visualized with C-DiGit^®^ Blot Scanner (LI-COR) by using the ECL Western Blotting Detection Reagents (Thermo Fisher Scientific, IL, USA). Image capture was carried out using image studio^®^ (LI-COR, version 5.0) software. Original western blots can be found at Appendix A.

### 2.11. ChIP

For chromatin immunoprecipitation (ChIP) experiments, the ChIP Assay Kit (Pierce Agarose ChIP Thermo Fisher Scientific) was used. Formaldehyde 1% was used to crosslink MM cells (1.5 × 10^7^), which were then lysed and sheared by sonication. Chromatin extracts were divided into equal amounts of immunoprecipitation with the MYC antibody (ab56), or rabbit IgG as a negative control (Santa Cruz Biotechnology), and then incubated on a rotator with ChIP Grade Protein A/G Plus Agarose for 3 h at 4 °C. Bound agarose beads were harvested by centrifugation (12,000 rpm, 15 s), and washed; incubation at 65 °C for 1.5 h with NaCl and proteinase K was then performed to revert cross-links and to obtain the precipitated protein–DNA complexes from washed beads.

ChIP products were purified using a Qiaquick kit (Qiagen), followed by qPCR using the following primer sets:

miR-22 (forward) 5′-ATGGTACCGAGGTCACACTTTC-3′

miR-22 (reverse) 5′-TTAAGCTTTCACCCTCCATCC-3′

Ch22 (forward) 5′-GGATGACAGGCATGAGGAATTA-3′

Ch22 (reverse) 5′-TGCTGCTTACTTGGGATATGAG-3′

## 3. Results

### 3.1. MYC–miR-22 Inverse Correlation Predicts Poor Response to IMiDs in MM Patients

To investigate the role of MYC-regulated miRNAs in MM, an in silico integrated [28] approach was performed. In particular, previously characterized MYC-repressed miRNAs [20] were matched to proprietary plasma cell (PC) dyscrasias datasets [27] to identify those miRNAs inversely correlated with MYC expression in MM patients. Importantly, miRNA profiling of our series of patient-derived specimens (n = 96 MM, n = 29 plasma cell leukemia (PCL) cases) disclosed a significant inverse correlation only between miR-22 and MYC expression in both MM and PCL cases, with an r value of −0.53 in the latter, demonstrating a clear correlation with disease progression (Figure 1a and Appendix A). Importantly, the inverse correlation between miR-22 and MYC levels were then validated by interrogating the public MMRF-CoMMpass dataset (774 MM patients) (Appendix A).

Immunomodulatory imide drugs (IMIDs) exert anti-MM activity by a direct cytotoxic effect and/or by triggering immune-based destruction of MM cells. Recent experimental evidence indicates that these effects are mediated by activation of CRBN-CRL4 E3 ubiquitin ligase, which, in turn, leads to IKZF1/IKZF3 degradation and, finally, MYC downregulation. On these bases, the potential role of MYC-driven miR-22 downregulation as a mediator of IMID-based treatment in MM patients was investigated by interrogating the MMRF-CoMMpass (IA15) dataset. Normalized MYC and MIR22HG expression levels were available for 774 BM_1 MM patients that were profiled by RNA sequencing (RNA-seq). The 774 MM samples of the entire RNA-seq dataset were stratified accordingly to MYC and MIR22HG expression levels, and the groups inversely combining the two extreme quartiles (Appendix A) were considered in relation to clinical outcome. In particular, 89 MM at higher MYC and lower MIR22HG expression (MYC_IV, MIR22HG_I quartiles) were compared to 69 MM at lower MYC and higher MIR22HG expression level (MYC_I, MIR22HG_IV quartiles) with respect to OS and PFS data. Notably, a significantly poorer survival rate was observed in PFS for those MM patients classified in MYC IV, MIR22HG I quartiles (Figure 1b, left), whereas no significant differences were observed between the two MYC–MIR22HG groups with respect to OS (data not shown).

Furthermore, we selected 353 MM patients that were treated with IMID-based therapies (alone, and in combination with bortezomib, carlfizomib, or both) as a first-line treatment and reached a completed regimen (270 patients) or developed disease progression/relapse (83 patients), respectively. A significantly lower MIR22HG expression level was observed in progressed/relapsed MM cases compared with those who reached an IMID-based completed regimen (Figure 1b, right), whereas similar MYC expression levels were observed in both groups. Notably, these results were confirmed in selected patients (340 MM) that were specifically treated with lenalidomide-based therapies (Figure 1c).

### 3.2. MYC Represses miR-22 Transcription in MM

To confirm the clinical findings, which suggested a potential important role of MYC-driven miR-22 regulation in MM pathogenesis and IMiD response, analysis of miR-22 levels was performed on a panel of MM cell lines. According to MYC expression, a global downregulation of miR-22 levels was found among MM cell lines, except for U266 cells, which instead were null for c-MYC [29] and displayed higher miR-22 levels as compared to other MM cell lines (Figure 2a). On the other hand, MYC-enforced expression by a lentivector in U266 cells (U266 MYC+) led to downregulation of miR-22, further suggesting a suppressive role of MYC on miR-22 expression (Figure 2b).

Next, different approaches were used to downregulate MYC in order to evaluate miR-22 modulation. First, transfection of MM cells with specific siRNAs was performed. We observed a strong reduction in MYC levels, which indeed occurred together with a significant increase in miR-22 (Appendix A). Then, two well-known small molecule MYC inhibitors were used. In particular, treatment of MM cells with 10058-F4 and JQ1 was able to induce a significant increase in miR-22 expression in MM cells (Figure 2c).

Finally, to confirm the direct transcriptional effect of MYC on miR-22 expression, CHIP analysis was performed. Importantly, a significant enrichment of MYC at miR-22 promoter in MM cells was found, thus confirming that MYC could repress miR-22 in MM by a direct “in cis” effect on its transcription (Figure 2d).

### 3.3. Enforced Expression of miR-22 Inhibits MYC Expression and Function

To fully elucidate the interaction between MYC and miR-22, a bioinformatic tool (TransmiR software, http://www.cuilab.cn/transmir, v2.0, accessed on 10 January 2021), which is based on 3730 literature-described transcription factors (TF)-miRNA functional interaction and 1,785,998 TF-miRNA interactions derived from ChIP-seq evidence, was used. Several transcription factors were predicted to bind to miR-22 promoter (Figure 3a). Interestingly, among them, this analysis predicted a feedback regulation with MYC. On these bases, a potential inhibitory role of miR-22 on MYC expression was also investigated.

To elucidate the molecular mechanisms involved in MYC downregulation induced by miR-22, MYC regulatory components were bioinformatically investigated, and then validated as a target of miR-22 (Figure 3b). This approach disclosed several downregulated genes, which are bona fide mediators of MYC downregulation induced by miR-22 in MM cells. In particular, among these are: (a) MYCBP and MAX, previously validated targets of miR-22, which act as a coactivators of MYC transcriptional activity; (b) IKZF3, which plays a pivotal role as a transcriptional activator of MYC expression; and (c) MYC itself, which has not been previously investigated as a target of miR-22. Consistently with this analysis, a strong downregulation of MYC mRNA and protein was found in MM cells transfected with miR-22 mimics as compared to miR-NC-transfected cells (Figure 3c,d), further suggesting a direct MYC regulation by miR-22, which required further investigation.

### 3.4. miR-22-Dependent Regulation of MYC Expression in MM Cells

Among MYC cofactors or MYC transcriptional activators, analysis of publicly available MM datasets showed a significant inverse correlation only between miR-22 expression and MYCBP, thus suggesting its pivotal role as potential mediator of miR-22-mediated MYC downregulation in MM cells (Figure 3a).

Indeed, analysis of public MM datasets showed that higher MYCBP mRNA levels correlated with poor overall survival in MM patients. Notably, MYCBP expression increased in plasma cells from MM patients as compared to healthy donors (N), confirming the pivotal role of this target in MM pathogenesis (Figure 4b).

To validate in silico findings, the effects on MYC expression after MYCBP knockdown was investigated. Importantly, transfection with MYCBP-specific siRNA led to strong MYC downregulation, thus recapitulating effects induced by miR-22 overexpression (Figure 4c). Next, to assess if the effect of miR-22 overexpression on MYC levels was actually mediated via MYCBP, AMO1 were co-transfected with miR-22 and a miR-22-insensitive MYCBP expression construct lacking 3′UTR. Our results showed that the non-targetable MYCBP partially rescued transfected cells from MYC downregulation, thus suggesting that other targets are also involved in the MYC–miR-22 loop in MM (Figure 4d).

### 3.5. miR-22 Triggers MYC-Dependent Synthetic Lethality

On these premises, a potential direct targeting of MYC mRNA was then investigated. In silico analysis by STarMir software (http://sfold.wadsworth.org, accessed on 20 January 2021) [30] identified four different predicted “seedless” target sites of miR-22 on MYC 3′-UTR, and one “seed” target site on MYC 5′-UTR (Appendix A). To validate these findings, a luciferase reporter assay was performed, using the target site with the highest prediction probability score. Luciferase activity was significantly reduced by miR-22 mimics in cells transfected with wild-type MYC reporter, while deletion of the miR-22 binding site in the MYC 3′-UTR abrogated the miR-22 suppression of luciferase activity, indicating that miR-22 directly downregulates MYC expression by binding to its regulatory regions (Figure 5a,b).

After identification of an MYC–miR-22 regulatory loop in MM, a mechanistic association between MYC downregulation and antiproliferative effects induced by miR-22 overexpression in MM cells was then investigated.

To test this hypothesis, antiproliferative effects of miR-22 on a panel of MM cell lines diverging for MYC expression were investigated. Notably, MYC-null U266 cells did not show significant cell proliferation impairment and apoptosis induction after miR-22 transfection, as compared to MYC-proficient MM cell lines (Figure 5c). Consistently, MYC overexpression by a lentivector (U266 MYC+) restored miR22-mediated killing as compared to empty vector transduced U226 cells (U266 EV) (Figure 5d).

Next, to assess if the effect of miR-22 on MM cell viability was actually mediated via MYC, AMO1 were co-transfected with miR-22 and a miR-22-insensitive MYC gene expression construct lacking 3′UTR and 5′UTR. Our results showed that the non-targetable MYC rescued transfected cells from MYC downregulation and apoptosis induced by miR-22 mimics (Figure 5e).

### 3.6. miR-22 Sensitizes MM Cells to Lenalidomide by Reducing MYC Expression

As mentioned above, IMiD activity is mediated by MYC degradation. On these premises, the effect of lenalidomide treatment on miR-22 expression was then investigated. To this aim, AMO1 and R8226 cells were treated with 10 µM of lenalidomide and, after 72 h, miR-22 levels were analyzed. Notably, a significant upregulation of miR-22 levels was found in MM-treated cells as compared to vehicle control, ranging from a two- to threefold increase (Figure 6a). To evaluate if this effect was actually mediated by MYC downregulation, the same experiment was performed on U266 EV and on U266 MYC-overexpressing cells. Interestingly, treatment with lenalidomide led to miR-22 upregulation only in U266 MYC cells, thus suggesting a pivotal role exerted by MYC in mediating miR-22 expression in MM cells upon lenalidomide treatment (Appendix A).

Since lenalidomide treatment led to miR-22 increase in MM cells, the effect on cell viability of miR-22 transfection combined with lenalidomide was next investigated. To enhance the translational relevance of our findings, this experiment was performed on bortezomib-resistant cells (ABZB), given the pivotal role exerted by MYC in bortezomib resistance acquirement [31]. In particular, ABZB cells were transfected with miR-22 and, after 24 h, were treated with lenalidomide 10 µM or vehicle. Importantly, combination treatment induced significantly higher MM cell death as compared to lenalidomide or miR-22 alone (Figure 6b, Appendix A). Interestingly, the antiproliferative effect induced by combination treatment occurred together with a strong downregulation of MYC in MM cells, thus confirming the relevance of MYC downregulation as a potential mediator of miR-22/lenalidomide combinatorial activity on MM cell viability (Figure 6b, right). Notably, similar results are found in primary relapsed MM cells (Appendix A). Conversely, transfection of MM cells with miR-22 inhibitor led to an increase in MYC expression, which rendered MM cells more resistant to lenalidomide (Figure 6c).

Next, a similar experiment was conducted on OPM2 lenalidomide-resistant cells (OPM2LenRes) [32]. Importantly, miR-22 overexpression restored lenalidomide sensitivity in OPM2LenRes by inducing MYC downregulation, further suggesting that this strategy represents a promising approach to restore IMiD sensitivity in MM refractory/relapsed patients (Figure 6d).

### 3.7. miR-22 Potentiates NK-Mediated Cytotoxicity Induced by Lenalidomide

IMiDs exert their immunomodulatory activity on MM cells also by activating NK-mediated cytotoxic activity [33]. Indeed, recent reports indicate that IMiDs upregulate the expression of the NKG2D- and DNAM-1-activating receptor ligands by reducing transcriptional inhibitory activity of the IKZF1/3-IRF4-MYC axis [34,35].

To investigate whether miR-22 repression of MYC signaling could also increase susceptibility to NK-mediated destruction of MM cells, NK ligand expression was evaluated by flow cytometry after miR-22 overexpression. Interestingly, a significant increase in NK-activating ligands, such as MICA/B and ULBP, was found. Conversely, a strong downregulation of NK negative regulator CD47 was observed (Figure 7a).

Then, to investigate if these phenotype changes translated into increased susceptibility to NK-mediated lysis of MM cells, cell cytotoxicity experiments were carried out. ABZB cells were transfected with miR-22 or scrambled control. After 24h, transfected cells were cocultured with human healthy donors-derived PBMCs at different effector/target ratios, and LDH release assay was then performed. Importantly, miR-22 overexpression led to higher in vitro cytotoxicity in AMO1 cells as compared to scrambled control (Figure 7b). Next, the capability of miR-22 to sensitize to lenalidomide NK activation against MM cells was investigated. In particular, ABZB cells were transfected with miR-22 and, after 24h, were treated with lenalidomide 10 µM or vehicle. Then, MM cells were cocultured with peripheral blood mononuclear cells in an effector–target ratio of 10:1, and NK activity was evaluated by degranulation assay. Notably, miR-22/lenalidomide combination induced significant NK degranulation, which translated into a twofold increase in CD107+ NK cells in the combination group as compared to single-agent treatment (Figure 7c,d); conversely, no synergistic effects were observed after transfection of MM cells with miR-22 inhibitor, which indeed did not induce significant modulations of NK ligands (Appendix A). Overall, these data indicate that MYC downregulation induced by miR-22 is able to potentiate both lenalidomide direct and NK-mediated cytotoxicity in MM cells.

## 4. Discussion

MYC aberrations are strongly involved in MM pathogenesis and correlate with poor clinical outcome and drug resistance [31,36]. MYC plays a critical role in the control of miRNAs expression and, in turn, MYC-regulated miRNAs influence all MYC-driven hallmarks of cancers [37].

In this work, we aimed to investigate the role of MYC-driven downregulation of miRNAs as a mediator of IMiD response. In this light, integrated bioinformatics analysis of miRNAs and MYC mRNA expression profiles indicated the existence of a miR-22–MYC regulatory loop. Indeed, by interrogating public, as well as proprietary MM gene expression profiles, we showed an inverse correlation between MYC and miR-22 expression. Consistently, MYC knockdown or its pharmacological inhibition led to a significant increase in miR-22 levels in MM cells. Thoroughly examining our findings, we confirm that MYC binds to miR-22 promoter [20], supporting our hypothesis of a negative regulation of MYC on miR-22 transcription in MM cells.

In a previous work, we provided evidence that miR-22 acts as a tumor suppressor in MM cells in vitro and in vivo. Importantly, we reported a significant downregulation of miR-22 in MM and PCL patients as compared to healthy subjects, as well as in MM cell lines, suggesting a critical tumor-suppressing role of this miRNAs in MM onset and progression [38]. Here, we reported the relevance of miR-22 as an MYC negative regulator in MM patients. Consistently, in this study, we highlighted the role of miR-22 as a bona fide endogenous MYC signaling inhibitor in MM, since its overexpression led to downregulation of, above MYC itself, several MYC cofactors, such MYCBP [39] and MAX [40], as well as MYC transcription activator, IKZF3. These data could be of translational relevance, taking into account that MYC signaling is overexpressed in more than half of MM tumors, and, currently, no specific MYC inhibitors are clinically available.

Importantly, we demonstrated that miR-22 exerted its tumor growth inhibitory activity mostly in MYC-proficient cells, further providing evidence of a miR22/MYC axis as a viable and attractive therapeutic target in MM.

Notably, we showed that lenalidomide, which constitutes the backbone of MM therapy and maintenance [40,41], led to a strong increase in miR-22, which occurred together with MYC downregulation in agreement with our hypothesis. On this basis, we indeed demonstrated that miR-22 and lenalidomide combination exerts synergistic antiproliferative activity, together with their combined effects on MYC, offering proof-of-concept for a new therapeutic approach, especially in IMiD-refractory MM patients. Indeed, by interrogating the largest available MM RNA-seq datasets (MMRF-CoMMpass), we found that patients with high miR-22 and low MYC levels have a better PFS when treated with IMIDs. Furthermore, these patients had a lower risk of relapse after IMID-based treatment, further confirming that higher miR-22 expression can empower IMID activity on MM cells. Moreover, our data could be consistent with the synergistic activity observed in MM and mantle cell lymphoma preclinical models, between BET bromodomain inhibitor (MYC transcriptional inhibitor) and lenalidomide [42,43].

Next, our attention moved to explore perturbations of miR-22 on other MYC-regulated hallmarks in MM cells. In fact, above induction of the pro-survival and proliferation gene, MYC controls the expression of different immune checkpoint proteins on the tumor cell surface, such as the innate immune regulator, CD47 (cluster of differentiation 47), and NK-activating ligands, such as MICA/B and PVR [43,44,45]. Indeed, it is well known that MYC downregulation enhances the antitumor immune response by inducing antitumor immune-mediated activity. Consistently, our findings suggest that MYC downregulation by miR-22 increases NK-activating ligands on the MM cell surface, such as MICA/B and ULBP 2/5/6, and a downregulation of NK-inhibiting ligands, such as CD47. Importantly, we showed that enforced expression of miR-22 sensitizes MM cells to lenalidomide NK-mediated cytotoxicity, thus indicating that miR-22 triggers both direct and immune-mediated tumor suppressor activity in MM cells.

The introduction of immune-modulatory drugs and proteasome inhibitors, as well as the use of novel monoclonal antibodies, in the therapeutic setting of MM has resulted in prolonged survival of patients. However, MM remains an incurable disease, and there is an urgent need for new therapeutic strategies, especially in patients with relapsed or refractory disease. In this context, our in silico and in vitro findings suggest that combination of lenalidomide with miR-22 mimics could represent a promising therapeutic strategy to target MYC addiction of MM cells.

Based on these findings, we hypothesize that MYC-proficient MM cells downregulate miR-22 levels to amplify MYC-driven myelomagenesis by promoting cell survival and escape from NK cytotoxic activity. However, at the same time, these events make MM cells more vulnerable to increased expression of miR-22, which indeed triggers MYC-dependent synthetic lethality. Notably, these findings are of translational relevance, since they could suggest miR-22 levels as a potential predictive factor to select patients who might better respond to IMiD-based therapy.

## 5. Conclusions

In summary, we uncover a miR22–MYC signaling loop in MM, in which miR-22 functions as a pivotal antitumor gatekeeper by direct cytotoxic activity and/or by immune-mediated activity. We think that our study together highlights the complexity and functional relevance of miR-22-associated gene regulation and signaling pathways in MM, and provides novel insights into MYCdriven tumorigenesis and IMiDs’ mechanism of action.

## Figures and Tables

**Figure 1 cancers-13-04365-f001:**
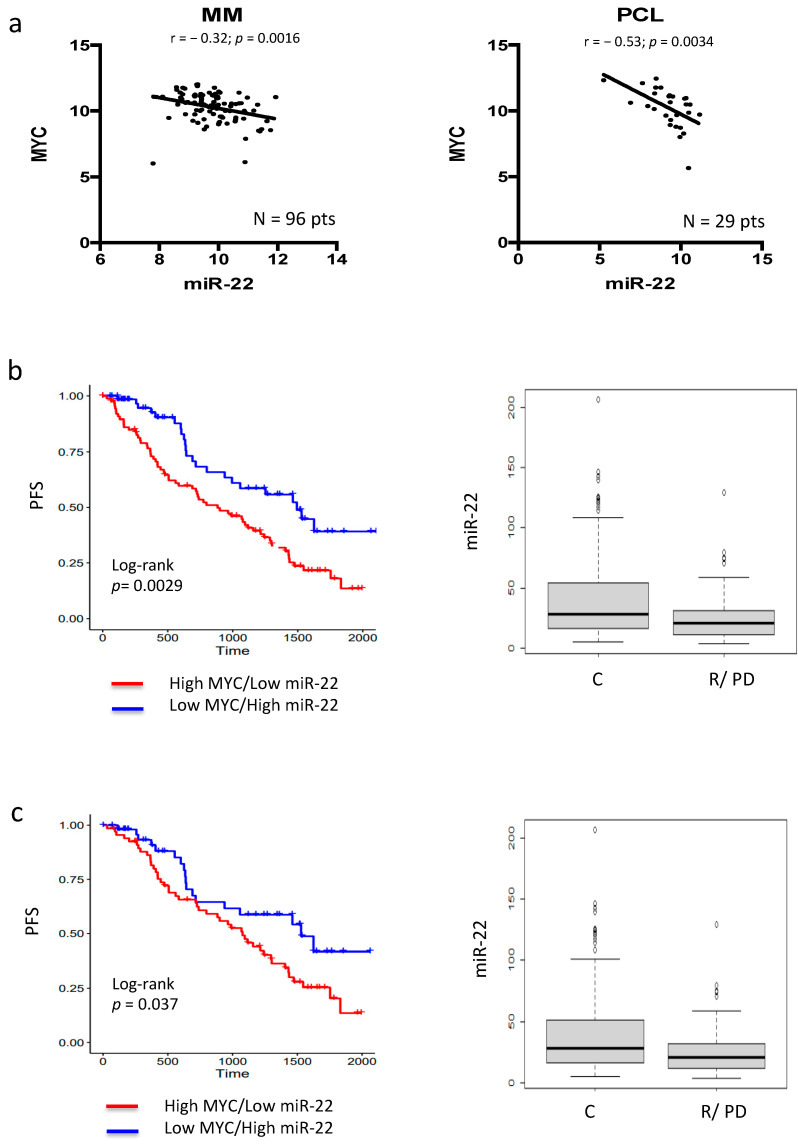
In silico prediction of MYC–miR-22 regulation in MM. (**a**) Graphs of correlations between endogenous mRNA expression levels of MYC and miR-22-3p in patients’ MM and PCL cases from the proprietary dataset (GSE73454 and GSE70254). (**b**) Data obtained from the MMRF-CoMMpass dataset. Left: progression-free survival (PFS) in IMID-treated MM patients of higher MYC and lower MIR22HG expression (MYC_IV, MIR22HG_I quartiles) as compared to lower MYC and higher MIR22HG expression (MYC_I, MIR22HG_IV quartiles). Right: MIR22HG expression in MM patients who completed regimen (R) or developed disease progression/relapse (R/PD) after IMID-based first-line treatment. (**c**) Data obtained from the MMRF-CoMMpass dataset. Left: progression-free survival (PFS) in lenalidomide-treated MM patients of higher MYC and lower MIR22HG expression (MYC_IV, MIR22HG_I quartiles) as compared to lower MYC and higher MIR22HG expression (MYC_I, MIR22HG_IV quartiles). Right: MIR22HG expression in MM patients who completed regimen (R) or developed disease progression/relapse (R/PD) after lenalidomide-based first-line treatment.

**Figure 2 cancers-13-04365-f002:**
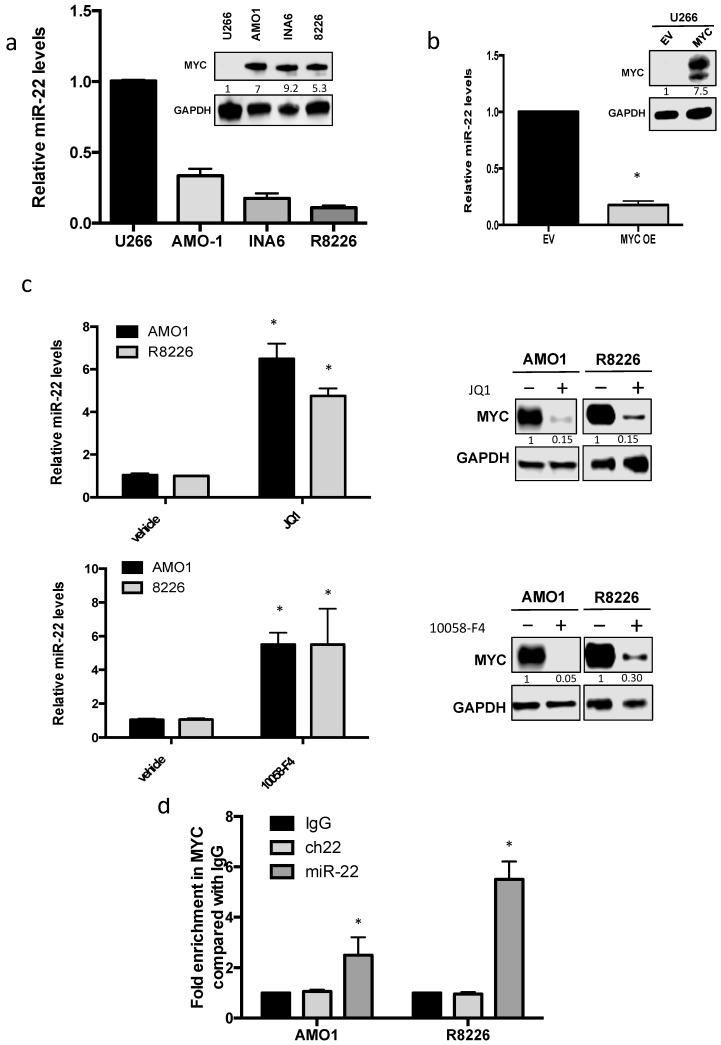
MYC downregulates miR-22 transcription in MM. (**a**) Left panel: miR-22 expression in indicated MM cell lines as evaluated by qRT-PCR. miR-22 levels in U266 cells were set as an internal reference. The results are shown as average miR-22 expression levels after normalization with RNU44 and ΔΔCt calculations. Right panel: immunoblot of MYC performed on plasma cells from MM cell lines. GAPDH was used as a loading control. (**b**) Left panel: miR-22 expression in U266 EV and U266 MYC as evaluated by qRT-PCR. The results are shown as average miR-22 expression levels after normalization with RNU44 and ΔΔCt calculations. Right panel: Western blot for MYC performed on U266 cells transfected with empty vector (EV) or MYC expressing plasmids. GAPDH was used as a loading control. (**c**) Top: AMO1 and RPMI-8226 cells were treated with JQ1 500 nM for 24 h. Left panel: miR-22 expression as evaluated by qRT-PCR. The results are shown as average miR-22 expression levels after normalization with RNU44 and ΔΔCt calculations. Right panel: immunoblot of MYC. GAPDH was used as a loading control. Bottom: AMO1 and RPMI-8226 cells were treated with 10058-F4 10 μM for 24 h. Left panel: miR-22 expression as evaluated by qRT-PCR. Right panel: immunoblot of MYC. GAPDH was used as a loading control. (**d**) qPCR for miR-22 promoter performed after ChIP with MYC antibody in AMO1 and R8226 compared with negative (ch22) and IgG controls. Results are average ± SD of three independent experiments performed in triplicate. * *p* < 0.05.

**Figure 3 cancers-13-04365-f003:**
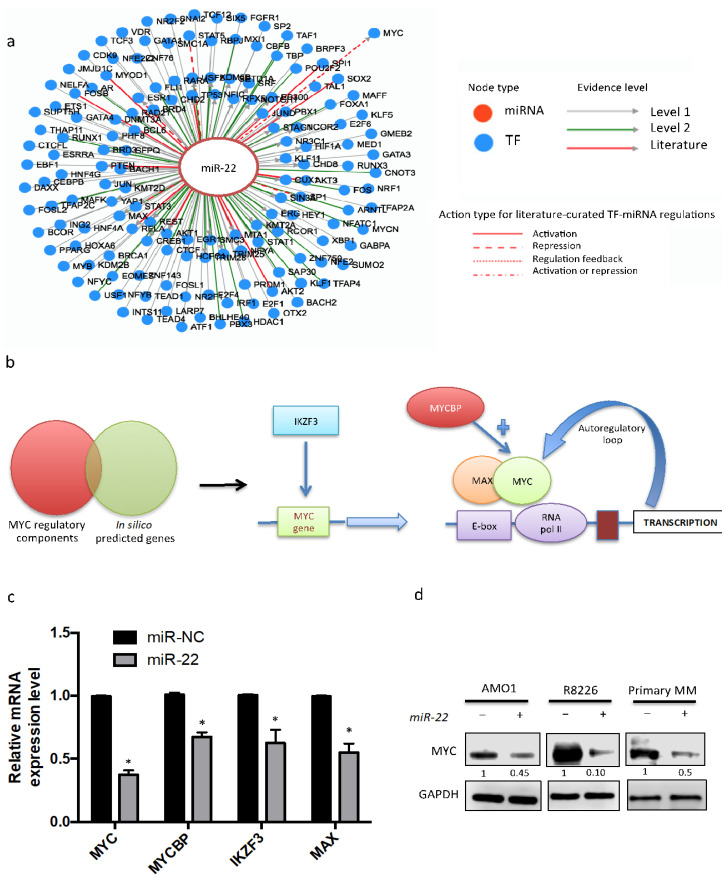
miR-22 counteracts MYC regulatory network in MM. (**a**) Data obtained from TransmiR software (http://www.cuilab.cn/transmir, v2.0, accessed on 10 January 2021) showing a feedback regulation of miR-22 by MYC. (**b**) Workflow of in silico integrated investigation performed to dissect miR-22-driven regulation of MYC expression. (**c**) qRT-PCR analysis of MYC regulatory network gene performed on AMO1 cells transfected with miR-22 or miR-NC. The results are average expression levels after normalization with GAPDH and ΔΔCt calculations. (**d**) Immunoblot of MYC performed on AMO1, R8226, and primary MM cells after 24 h from transfection with miR-22 or miR-NC. GAPDH was used as a loading control. Results are average ± SD of three independent experiments performed in triplicate. * *p* < 0.05.

**Figure 4 cancers-13-04365-f004:**
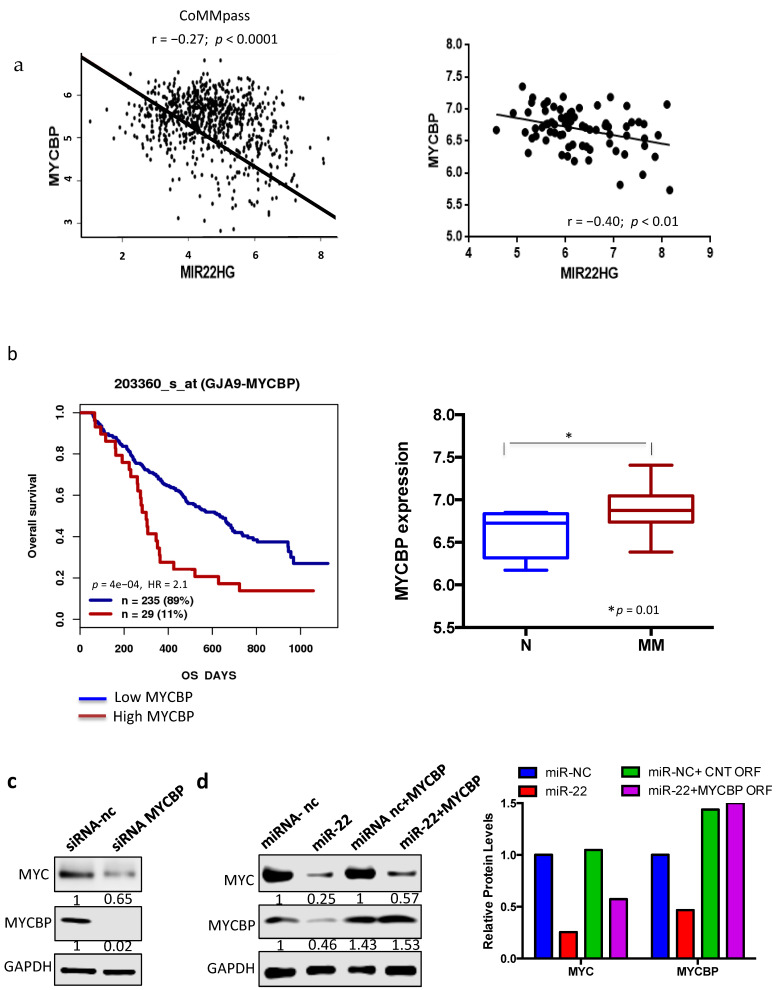
MYCBP as a potential mediator of the MYC–miR-22 loop. (**a**) Graphs of correlations between endogenous mRNA expression levels of MYCBP and MIR22HG in patients’ multiple myeloma cells from published MMRF-CoMMpass and GSE39925 datasets. (**b**) Left: prognostic relevance of MYCBP expression on overall survival (OS) of MM patients, resulting from GSE9782 dataset analysis. Right: MYCBP expression in MM patients as compared to healthy donors (N), resulting from GSE47552 dataset analysis. (**c**) AMO1 cells were transfected with scramble control or MYCBP-specific siRNA. Immunoblot of MYC and MYCBP were performed 48h after transfection. GAPDH was used as a loading control. (**d**) AMO1 cells were co-transfected with MYCBP ORF or control ORF and miR-22 or scrambled oligonucleotides. Immunoblot (**left**) and protein quantification analysis (**right**); GAPDH was used as a loading control. Analysis was performed 48 h after cell transfection. Results are average ±SD of three independent experiments performed in triplicate. * *p* < 0.05.

**Figure 5 cancers-13-04365-f005:**
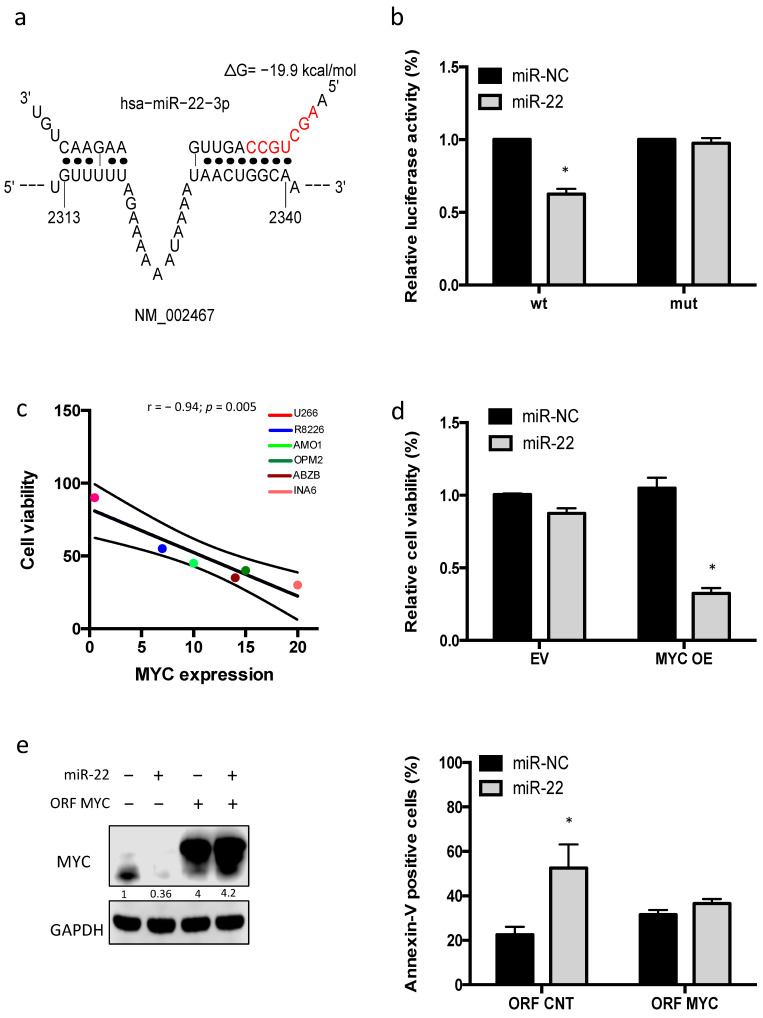
miR-22 as a direct negative regulator of MYC in MM. (**a**) Predicted binding site of miR-22 on MYC 3′UTR. (**b**) Dual luciferase assay performed on AMO1 cells co-transfected with firefly luciferase constructs containing the 3′ UTR of MYC or a deletion mutant (3′ UTR mt) and miR-22 or miR-NC, as indicated. The data are shown as relative luciferase activity of miR-22-transfected cells compared with the control (miR-NC). (**c**) Pearson correlation between the MYC expression (*x*-axis) and viability relative to miR-NC (miR-22, *y*-axis), evaluated 96 h from transfection. The linear regression line is plotted in black. Each dot represents one MM cell line. (**d**) Cell viability was evaluated 96 h after transfection of, respectively, U266 EV and U266 MYC+ with miR-22 mimics or scrambled controls (miR-NC). (**e**) AMO1 cells were co-transfected with MYC ORF or control ORF and miR-22 mimics or miR-NC. Left panel: immunoblot shows the levels of MYC and GAPDH 48 h after cell transfection. Right panel: annexin V assay was performed 48 h after transfection. Results are average ± SD of three independent experiments performed in triplicate. * *p* < 0.05.

**Figure 6 cancers-13-04365-f006:**
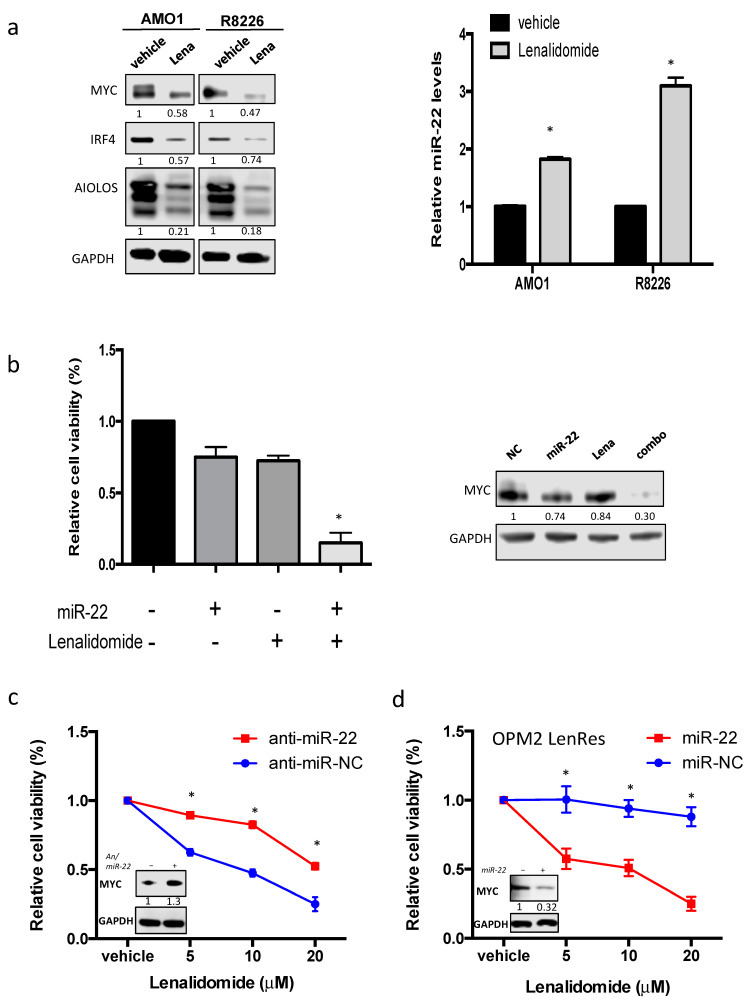
miR-22 modulates lenalidomide sensitivity of MM cells. (**a**) AMO1 and RPMI-8226 cells were treated with vehicle or lenalidomide 10 μM for 72 h. Left panel: immunoblot of MYC, IRF4, and AIOLOS. GAPDH was used as a loading control. Right panel: miR-22 expression as evaluated by qRT-PCR. The results are shown as average miR-22 expression levels after normalization with RNU44 and ΔΔCt calculations. (**b**,**c**) ABZB cells were transfected with miR-22 (**b**) or anti-miR-22 (**c**) and, after 24 h, were treated with lenalidomide 10 μM or vehicle. Cell viability was evaluated after 72 h from lenalidomide treatment. Immunoblot of MYC was performed 48 h from Lenalidomide treatment. GAPDH was used as a loading control. (**d**) OPM2 LenRes cells were transfected with miR-22 and, after 24 h, were treated with lenalidomide 10 μM or vehicle. Cell viability was evaluated after 72 h from lenalidomide treatment. Immunoblot of MYC was performed 48 h from Lenalidomide treatment. GAPDH was used as a loading control. Results are average ±SD of three independent experiments performed in triplicate. * *p* < 0.05.

**Figure 7 cancers-13-04365-f007:**
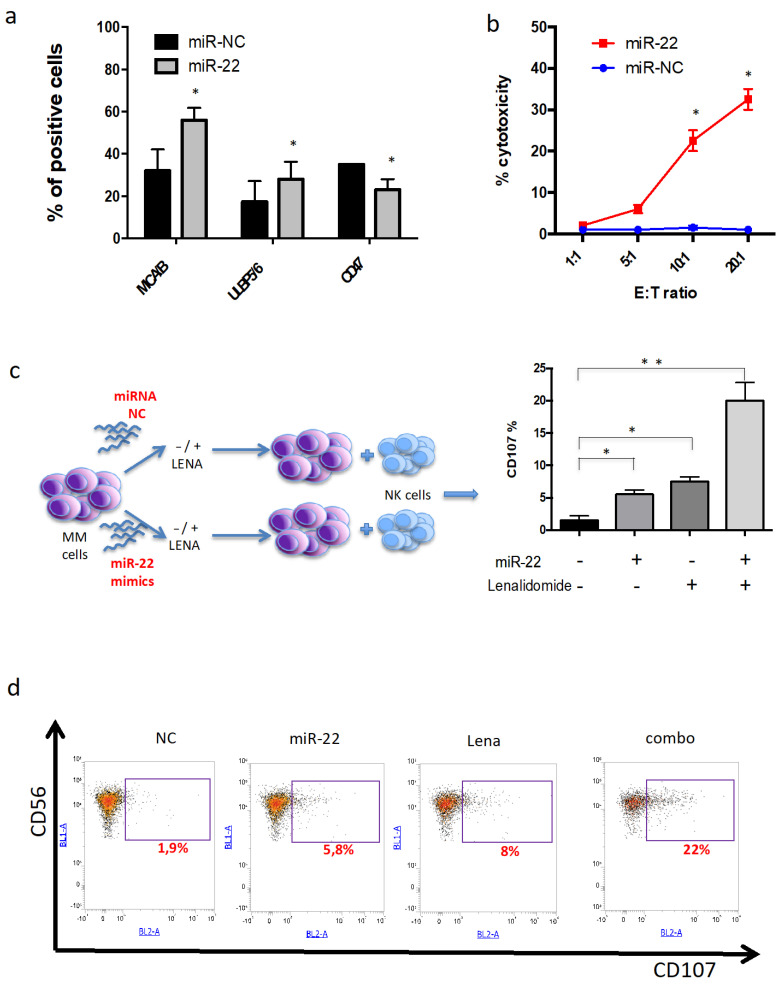
miR-22 overexpression sensitizes MM cells to NK killing. (**a**) Flow cytometry of representative NK-activating and -inhibiting ligands performed on ABZB cells transfected with miR-22 or miR-NC. (**b**) Percentage of cytotoxicity monitored by LDH release assay on ABZB cells transfected with miR-22 or miR-NC and cocultured with PBMCs at different E:T ratio for 48 h. (**c**) Workflow of degranulation assay and percentage of CD107a positivity in effector cells (CD56+ NK cells) cocultured with ABZB, after co-treatment with miR-22 and lenalidomide. (**d**) Representative FACS traces of degranulation assay showing synergistic CD107a increase upon miR-22 and lenalidomide combination. Results are average ± SD of three independent experiments performed in triplicate. * *p* < 0.05; ** *p* < 0.01.

## Data Availability

The data presented in this study are available in this article (and Appendix A).

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
