# Peer review of "miR-22 Modulates Lenalidomide Activity by Counteracting MYC Addiction in Multiple Myeloma"

_cancers, 2021, doi:10.3390/cancers13174365_

Round 1
Reviewer 1 Report
In the manuscript by Caracciolo et al. the authors reported that MYC directly represses the transcription of tumor-suppressor miR-22 in MM and that low miR-22 expression is associated with IMiDs resistance in MM patients. Furthermore, they showed that Lenalidomide treatment increased miR-22 activity by antagonizing MYC inhibitory effect and that the combination of Lenalidomide with miR-22 mimics resulted in synergistic anti-MM activity.
However, I have some minor comments:
- In Figure 1, the authors showed an inverse correlation between MYC and miR-22 in patient's MM and PCL cases from the proprietary dataset. Could the authors validate this finding in larger public MM datasets, such as MMRF-CoMMpass?
- In Figure 6, the authors showed that Lenalidomide treatment increases miR-22 expression in AMO1 and R8226 cells. To evaluate if this effect was actually mediated by MYC down-regulation, the authors could perform the same experiment on U266 EV and on U266 MYC overexpressing cells.
- In Abstract, please change the sentence, “MYC hyper-activation is involved in development of Immunomodulatory imide drugs (IMiDs) resistance ----” to “MYC hyper-activation is involved in development of resistance to immunomodulatory imide drugs (IMiDs)”.
- In Methods section, please correct “CellLysis Buffer” to “Cell Lysis Buffer”.
- Throughout manuscript, please remove unnecessary capitalizations. For example, correct “poor Overall Survival in MM patients” to “poor overall survival in MM patients”.
Author Response
We thank the Reviewers for all criticisms and detailed suggestions which will significantly improve our manuscript.
Reviewer 1
In the manuscript by Caracciolo et al. the authors reported that MYC directly represses the transcription of tumor-suppressor miR-22 in MM and that low miR-22 expression is associated with IMiDs resistance in MM patients. Furthermore, they showed that Lenalidomide treatment increased miR-22 activity by antagonizing MYC inhibitory effect and that the combination of Lenalidomide with miR-22 mimics resulted in synergistic anti-MM activity.
However, I have some minor comments:
- In Figure 1, the authors showed an inverse correlation between MYC and miR-22 in patient's MM and PCL cases from the proprietary dataset. Could the authors validate this finding in larger public MM datasets, such as MMRF-CoMMpass?
We thank the Reviewer for the careful evaluation of our manuscript and for this relevant suggestion. Accordingly, in the present version, we have included a graph describing the inverse correlation between MYC and miR-22 as evaluated in MMRF-CoMMpass dataset, in the Supplementary Figure 1b. Moreover, we have amended the text to describe this finding, which confirm our initial analysis (page 6).
- In Figure 6, the authors showed that Lenalidomide treatment increases miR-22 expression in AMO1 and R8226 cells. To evaluate if this effect was actually mediated by MYC down-regulation, the authors could perform the same experiment on U266 EV and on U266 MYC overexpressing cells.
We completely agree with the Reviewer on this important point. Consistently, we performed the proposed experiments on U266 EV and on U266 MYC overexpressing cells. As shown in new Supplementary Figure 4a, up-regulation of miR-22 was observed in MYC ON condition only. Moreover, we have amended the text to describe this result, which supports our proposed mechanism (page 17).
- In Abstract, please change the sentence, “MYC hyper-activation is involved in development of Immunomodulatory imide drugs (IMiDs) resistance ----” to “MYC hyper-activation is involved in development of resistance to immunomodulatory imide drugs (IMiDs)”.
- In Methods section, please correct “CellLysis Buffer” to “Cell Lysis Buffer”.
- Throughout manuscript, please remove unnecessary capitalizations. For example, correct “poor Overall Survival in MM patients” to “poor overall survival in MM patients”.
We thank Reviewer 1 for the careful reading of our manuscript and for raising all these suggestions. Accordingly, we have amended the text to avoid confusion and improve paper readability

Reviewer 2 Report
In their paper, Caracciolo, Riillo et al demonstrate the role of miR-22 in multiple myeloma. In their previous work, they identified miR-22 as negative regulator of LIG3 in MM. Increased level of miR-22 increased DNA damage inhibiting MM cell growth in vitro and in vivo. In current paper, they focused on the miR-22-MYC axis.
Introduction is well-written and presents the background of the study. Methods are adequatly described. Authors demonstrated the inverse correlation between MYC and miR-22 and they found that it correlated with poor response to IMIDs. Further, they demonstrated an axis between miR-22 and MYC and direct binding of MYC to miR-22 promoter.
In general, study is very well design and presents interesting results. I congratulate the authors. I have only a few comments that will improve the work.
Major points:
1. What are the effects of miR-22 inhibition in U266 cells?
2. Fig. 4b - MYCBP mRNA levels is not visible on this graph.
3. Add the number of biological replicates of the data presented in the figures to legends.
4. Is Fig4c, MYCBP densitometry correct? There is no expression of MYCBP based on immunoblot, however, protein quantification says 0.15
Author Response
We thank the Reviewers for all criticisms and detailed suggestions which will significantly improve our manuscript.
Reviewer 2
In their paper, Caracciolo, Riillo et al demonstrate the role of miR-22 in multiple myeloma. In their previous work, they identified miR-22 as negative regulator of LIG3 in MM. Increased level of miR-22 increased DNA damage inhibiting MM cell growth in vitro and in vivo. In current paper, they focused on the miR-22-MYC axis.
Introduction is well-written and presents the background of the study. Methods are adequatly described. Authors demonstrated the inverse correlation between MYC and miR-22 and they found that it correlated with poor response to IMIDs. Further, they demonstrated an axis between miR-22 and MYC and direct binding of MYC to miR-22 promoter.
In general, study is very well design and presents interesting results. I congratulate the authors. I have only a few comments that will improve the work.
Major points:
- What are the effects of miR-22 inhibition in U266 cells?
We thank Reviewer 2 for the careful evaluation of our manuscript. Inhibition of miR-22 did not induce significant effects on U266 cell viability (see attached figure).
- Fig. 4b - MYCBP mRNA levels is not visible on this graph.
We thank again the Reviewer for this suggestion. Accordingly, in the revised version of the manuscript, we have modified Fig.4b to improve its readability.
- Add the number of biological replicates of the data presented in the figures to legends.
As suggested, we have now included in the legends the number of biological replicates of the data presented in the figures.
- Is Fig4c, MYCBP densitometry correct? There is no expression of MYCBP based on immunoblot, however, protein quantification says 0.15.
We thank the Reviewer for this important observation. We have analyzed again and accordingly corrected protein quantification of Fig.4c, to avoid confusion and improve its readability.
